# 1,5-Benzothiazepine Derivatives: Green Synthesis, In Silico and In Vitro Evaluation as Anticancer Agents

**DOI:** 10.3390/molecules27123757

**Published:** 2022-06-10

**Authors:** Michelyne Haroun, Santosh S. Chobe, Rajasekhar Reddy Alavala, Savita M. Mathure, Risy Namratha Jamullamudi, Charushila K. Nerkar, Vijay Kumar Gugulothu, Christophe Tratrat, Mohammed Monirul Islam, Katharigatta N. Venugopala, Mohammed Habeebuddin, Mallikarjun Telsang, Nagaraja Sreeharsha, Md. Khalid Anwer

**Affiliations:** 1Department of Pharmaceutical Sciences, College of Clinical Pharmacy, King Faisal University, Al-Hofuf 31982, Al-Ahsa, Saudi Arabia; ctratrat@kfu.edu.sa (C.T.); kvenugopala@kfu.edu.sa (K.N.V.); 2MGV’s L.V.H. Arts, Science and Commerce College, Panchavati, Nashik 422009, India; chobess222@gmail.com; 3Shobhaben Pratapbhai Patel School of Pharmacy & Technology Management, SVKM’s NMIMS, V.L. Mehta Road, Vile Parle (W), Mumbai 400056, India; 4MGV’s M. S. G. Arts, Science and Commerce College, Malegaon-Camp, Tal-Malegaon, Nashik 423203, India; savita1190@gmail.com; 5Koneru Lakshmaiah Education Foundation, K L College of Pharmacy, Guntur 522502, India; r.namratha747@gmail.com; 6MGV’s Arts, Science and Commerce College, Manmad 422009, India; cknerkar@gmail.com; 7Janagoan Institute of Pharmaceutical Sciences, Janagoan 506167, India; vkgugulothu@gmail.com; 8Department of Biomedical Sciences, College of Clinical Pharmacy, King Faisal University, Al-Hofuf 31982, Al-Ahsa, Saudi Arabia; mislam@kfu.edu.sa; 9Department of Biotechnology and Food Science, Faculty of Applied Sciences, Durban University of Technology, Durban 4000, South Africa; 10Department of Biomedical Sciences, College of Medicine, King Faisal University, Al-Hofuf 31982, Al-Ahsa, Saudi Arabia; hmohammed@kfu.edu.sa; 11Department of Surgery, College of Medicine, King Faisal University, Al-Hofuf 31982, Al-Ahsa, Saudi Arabia; mvtelsang@kfu.edu.sa; 12Department of Pharmaceutics, Vidya Siri College of Pharmacy, Off Sarjapura Road, Bangalore 560035, India; 13Department of Pharmaceutics, College of Pharmacy, Prince Sattam Bin Abdulaziz University, Al-Alkharj 11942, Saudi Arabia; m.anwer@psau.edu.sa

**Keywords:** 1,5-benzothiazepines, PEG-400, molecular docking, cytotoxicity, anticancer

## Abstract

Considering the importance of benzothiazepine pharmacophore, an attempt was carried out to synthesize novel 1,5-benzothiazepine derivatives using polyethylene glycol-400 (PEG-400)-mediated pathways. Initially, different chalcones were synthesized and then subjected to a cyclization step with benzothiazepine in the presence of bleaching clay and PEG-400. PEG-400-mediated synthesis resulted in a yield of more than 95% in less than an hour of reaction time. Synthesized compounds 2a–2j were investigated for their in vitro cytotoxic activity. Moreover, the same compounds were subjected to systematic in silico screening for the identification of target proteins such as human adenosine kinase, glycogen synthase kinase-3β, and human mitogen-activated protein kinase 1. The compounds showed promising results in cytotoxicity assays; among the tested compounds, 2c showed the most potent cytotoxic activity in the liver cancer cell line Hep G-2, with an IC_50_ of 3.29 ± 0.15 µM, whereas the standard drug IC_50_ was 4.68 ± 0.17 µM. In the prostate cancer cell line DU-145, the compounds displayed IC_50_ ranges of 15.42 ± 0.16 to 41.34 ± 0.12 µM, while the standard drug had an IC_50_ of 21.96 ± 0.15 µM. In terms of structural insights, the halogenated phenyl substitution on the second position of benzothiazepine was found to significantly improve the biological activity. This characteristic feature is supported by the binding patterns on the selected target proteins in docking simulations. In this study, 1,5-benzothiazepines have been identified as potential anticancer agents which can be further exploited for the development of more potent derivatives.

## 1. Introduction

In synthetic medicinal chemistry, there are several privileged structures with different functional groups that can be considered for a variety of biological activities; 1,5-benzothiazepines (BTZ), as part of one of such privileged scaffold, has been of immense significance to the field of medicinal chemistry. Currently, BTZs are among the most broadly used drugs in the treatment of cardiovascular disorders, including examples such as Diltiazem, Thiazesim, and Clentiazem [1]. 

BTZ derivatives have been found to have activity against different target proteins, and are of particular attention for lead development [2]. The BTZ nucleus has been used as a cardiovascular modulator acting on several G-protein coupled receptors as an antagonist [3], such as the antiarrhythmic (CCK) receptor [4], Angiotensin-Converting Enzyme [5], Angiotensin II receptor [6], etc. Hemodynamic effects, anti-cancer activity [7], and spasmolytic activity [8,9,10], as well as anti-ulcer activity [11], have recently been reported, as had a central nervous system depressant effect [12]. The inhibition of the tyrosine kinase epidemic receptor of growth [13], which is related to the stabilization of the FKBP12 complex skeletal muscle channel–ryanodine receptor, has been reported. Phase II clinical trials on antiarrhythmic antihypertensive calcium (Ca^2+^) channel antagonistic activity are being carried out on two spinoffs, one of which is 7-bromo-3(S)-butyl-3-ethyl-8-hydroxy-5-phenyl-2,3,4,5-tetrahydro-1,5-benzothiazepine-1,1- dioxide (GW-577). Several BTZs are currently used in clinical practice; one of the most widely used classes is cardiovascular drugs such as diltiazem and clentiazem (Figure 1). As part of our ongoing efforts to identify new chemical entities (NCEs) endowed with biological activity, we considered the possibility of using a novel combination approach to the BTZ scaffold in order to investigate its anti-cancer properties. 

Numerous efforts have been undertaken in recent years for the development of new synthetic methodologies using renewable energy resources for shifting society away from non-renewable resources to environmentally-friendly biomass [14,15,16,17,18]. Organic molecules are key to this tedious process of drug discovery. In particular, there is a growing field of medicinal chemistry research that progresses heterocyclic materials; 1,5-derivatives of BTZ, such as those previously discovered to be drug candidates, have always drawn special interest [19,20,21]. In the current investigation, we employ an efficient green chemical approach for the synthesis of titled compounds, using an eco-friendly polyethylene glycol 400 (PEG-400) derivative as the reaction media [22]. The reaction involves the cyclic condensation of *o*-amino thiophenol (2) with chalcones (1) (Figure 2), which is a clean and environmentally friendly method for the synthesis of BTZs. The synthesized compounds were then evaluated for their in vitro anticancer properties. All the designed compounds were subjected to in silico screening for the identification of binding interactions with known targets, which are all potential key macromolecules in disease progression.

## 2. Results and Discussion

### 2.1. Synthesis of 2,3-Dihydro-1,5-benzothiazepines

We attempted to design a mild and efficient process for the synthesis of BTZ derivatives in the presence of PEG-400 and bleaching earth clay support (Figure 2). [23,24,25,26] Bleaching earth clay (pH 12.5) (BEC) was used in the current study, as it is an efficient heterogeneous catalyst for the synthesis of α, β-unsaturated carbonyl compounds thanks to its particle size (5 µm) and large surface area. It can be effectively used for the synthesis of base-catalyzed reactions to improve yields and reaction times. The activated form of BEC (10M) was used, regenerated after each cycle of reaction, and then used again for the next cycle of the reaction, and the same was done for PEG-400. There were no significant differences observed in the properties of BEC, and it was used for several runs without any effect on the outcome. 

The initial step deals with PEG-400-mediated condensation of substituted chalcones (1a–j) with 2-amino-4-methylbenzenethiol to afford the cyclized products 2a–j (scheme shown in Figure 2). Different reaction conditions, such as temperature, solvent, and time were investigated in order to identify the optimum condition for the synthesis of BTZ. It was noted that the reaction failed to occur at room temperature. Next, we carried out reactions at elevated temperatures and found that 60 °C is the ideal temperature to afford the derivative in high yields. Moreover, the reaction time was found to decrease to 55 min. Product formation was observed from the temperature of 40 °C, reached a maximum yield at 60 °C, after which the yield started to decline when applying higher temperatures (Table 1) and there was a notable reduction in reaction completion time. To compare the efficiency of the PEG-400, parallel reactions were carried out with various solvents such as dichloromethane, ethanol, and acetonitrile under the same reaction conditions, and differences in yield and reaction completion time were noticed (Table 2 and Figure 3). Among the solvents, PEG-400 was found to provide the maximum synthetic yield in less than an hour of reaction time, whereas the use of conventional solvents resulted in lower yields with extended reaction process times (around 4 h). Furthermore, the PEG-400 we used was recycled and subjected to reapplication for the next cycle of reaction, and was found to be useful even after four runs without any loss of activity (Figure 4). The same may not be possible with the other conventional solvents, even though they are recyclable in industry setups. By employing the optimized reaction conditions, the remaining designed compounds were synthesized and subjected to physicochemical and spectral studies. A total of ten molecules were synthesized with different substitutions on the 4′-Phenyl group.

The IR spectra of the synthesized compounds showed characteristic absorption bands in the region of 3060–3040 cm^−1^ because of the Ar–C-H of 1,5-benzothiazepine. The absorption band at 3150–3350 cm^−1^ resulted from–OH stretching at 1610–1590 cm^−1^ (C=N stretching). The C–Cl functional group resulted in a stretching band at 680–800 cm^−1^, while 600–700 cm^−1^ was caused by C–Br, appearing whenever it was present in one of the compounds. The absence of a sharp and intense absorption around 1690 cm^−1^ indicated the absence of the -C=O group of chalcone, confirming the formation of the cyclized products. The -C=N characteristic band for benzothiazepines was observed between 1610–1590 cm^−1^, unequivocally proving their structures. The ^1^H-NMR spectra revealed that HA, HB, HX, and the pattern of the 1,5-benzothiazepine ring could be seen as a doublet of doublets at δ 3.1–3.25, δ 3.43–3.61, δ 5.15–5.3 ppm, respectively, because of the two non-equivalent magnetically equivalent protons of the 1,5-benzothiazepine ring methylene group at position three. A single phenolic proton appeared near δ 10.5–12.50, corresponding to the phenolic–OH protons. All of the data on these peaks were in agreement with those reported in the literature. 

### 2.2. Biological Evaluation

#### Anti-Proliferative Activity

The synthesized pure benzothiazepine derivatives were explored for evaluation of in vitro cytotoxicity by MTT assay in two human cancer cell lines, liver (Hep-2) and prostate (DU-145), along with human embryonic liver cell lines (L02), for selective toxicity. The IC_50_ and percentage inhibition values of the tested compounds are presented in Table 3. It was found that the synthesized compounds are relatively non-toxic at 100 µM concentration in L02 cell lines, with a maximum of 52.16 ±1.42% inhibition by compound 2e, whereas the standard, Methotrexate, showed 78.23 ± 1.86% inhibition. It was observed that most of the tested compounds demonstrated considerable anti-proliferative activity against both tested cell lines. The compounds 2c, 2f, and 2j in particular exhibited high anticancer inhibition against both cell lines compared to the standard, methotrexate. The compounds possessing halogen substitution in 1′,4′-positions of the 2-phenyl ring on the benzothiazepine showed promising anticancer activities in the tested cell lines. The observed activity can be attributed to the activation of the ring by the electron-withdrawing nature of the halogen atoms. It was further observed in the docking studies of the compounds that 2c, 2f, and 2j demonstrated favorable hydrophobic binding interactions with the proteins. 

Among the tested compounds, 2c showed potent cytotoxic activity in the Hep G-2 (liver cancer) cell line, with an IC_50_ value of 3.29 ± 0.15 µM, whereas the standard drug IC_50_ was 4.68 ± 0.17 µM. In the same cell line, compounds 2f and 2j exhibited good activity, with IC_50_ values of 4.38 ± 0.11 and 4.77 ± 0.21 µM, respectively. The remaining tested compounds exhibited moderate activity, with the IC_50_ values of 3.29 ± 0.15 to 8.56 ± 0.14 µM, whereas in the prostate cancer cell line (DU-145) the compounds had IC_50_ values of 15.42 ± 0.16 to 41.34 ± 0.12 µM and the standard drug had an IC_50_ of 21.96 ± 0.15 µM. Among the tested compounds, compound 2j was observed to possess potent anti-proliferative properties, with an IC_50_ value of 15.42 ± 0.16 µM. The notable molecules in the series were 2c, 2f, and 2g, which showed good cytotoxic activity in the prostate cancer cell line. The incorporation of various halogen-substituted phenyl groups in the second position of benzothiazepine was crucial in maintaining good anticancer properties.

Three molecules 2c, 2f, and 2j of the present series of benzothiazepine presented promising cytotoxic activities against the tested cancer cell lines in comparison with the standard drug, Methotrexate. The observed anti-proliferative property could be attributed to the incorporation of halogen substituted phenyl group on the second position of the benzothiazepine nucleus, which may lead to favorable binding interaction with the target proteins responsible for improved chemotherapeutic activity. Nevertheless, further studies with judicious structural modifications of the benzothiazepine scaffold are needed in order to provide insight into the mechanism of action of the titled compounds.

### 2.3. Molecular Docking

PASS online activity prediction was performed in order to obtain information about the possible molecular targets of the active BTZ compounds. It was predicted that our BTZ analogues would have the ability to inhibit (Pa > 0.7) various kinase proteins such as human adenosine kinase, glycogen synthase kinase-3β, and human mitogen-activated protein kinase 1 enzymes, along with other prominent target proteins. Later, through mining the literature on BTZ derivatives and their activity profiles, it was found that the above kinases are potential target sites for numerous biological activities.

AutoDock 4.2, a widely distributed software for investigating molecular docking, was used to perform computer-simulated docking studies. An AutoDock study of BTZ was performed on the various kinase enzymes. As illustrated in Table 4 these AutoDock simulations provided the predicted binding free energy (ΔG_b_ kcal/mol) and inhibitory constants (Ki) for the respective kinase enzymes.

In the docking study against adenosine kinase (2I6B), the tested compounds showed binding energies between −7.65 and −2.68 kcal/mol, which confirms the potential binding affinity of these compounds for the binding cavity of the enzyme. The docking protocol was validated by performing a docking simulation of the drawn structure of the co-crystallized ligand (89I) and comparing it with the co-crystallized conformation present in the protein (Figure 5). The BTZ derivatives were found to possess inhibitory constants (Ki) in the range of 2.49 to 21.68 µM, whereas the value of the native ligand was 1.66 µM for the same enzyme (Table 4). The compounds interacted with the binding site residues Phe 201, Leu 40, Phe 170, Tyr 206, Asp 300, Thr 66, and Leu 138, showing both H-bonding and non-hydrogen bonding interactions (Figure 5 and Figure 6). Among the tested compounds, 2c and 2d, which contain iodine substitutions, activated the aromatic system and made strong interactions with the binding site residues. The H-bond length between the ligand and protein residues was found to be in the range of 1.93 to 2.85 Å. 

In the docking study against glycogen synthase kinase-3β (1Q41), for GSK-3β, the tested compounds were predicted to have binding energies between −5.27 and −0.04 kcal/mol. The BTZ derivatives were found to possess inhibitory constants (Ki) in the range of 11.67 to 36.05 µM, whereas for the native ligand, IXM, the value was 9.03 µM for this enzyme. The compounds were found to form interactions with the binding cavity residues Phe 209, Asn 195, Lys 97, Val 127, Met 143, Leu 197, Cys 207, and Asp 152, with both H-bonding and non-hydrogen bonding interactions (Figure 7). Among the tested compounds, 2d and 2f, which contain iodine substitutions, activated the aromatic system and formed strong interactions with the binding site residues. The H-bond length between the ligand and protein residues was found to be in the range of 1.84 to 2.64 Å. 

In the docking study against human mitogen-activated protein kinase 1 (3w8q), the tested compounds showed bonding energies between −11.71 and −0.35 kcal/mol, which confirms the potential binding affinity of these compounds for the binding cavity of the enzyme. The docking method was validated by performing a docking simulation of the drawn structure of the co-crystallized ligand (AGS) and compared with the co-crystallized conformation present in the protein (Figure 8). The BTZ derivatives were found to possess inhibitory constants (Ki) in the range of 33.09 to 76.97 µM, whereas for AGS the value was found to be 21.23 µM for this enzyme (Table 4). The compounds interacted with the binding site residues Lys 91, Gly 210, Gly 77, Val 211, Asn 78, Aer 212, Gly 79, Met 146, and Glu 144, with both H-bonding and non-hydrogen bonding interactions (Figure 8). Among the tested compounds, 2j and 2c, which contain iodine and bromine substitutions, respectively, activated the aromatic system and made strong interactions with the binding site residues. The H-bond length between the ligand and protein residues was found to be in the range of 1.65 to 2.92 Å.

Adenosine kinase inhibitors are known to possess anti-inflammatory, antinociceptive, and anticonvulsant activity in animal models. Recent reports on AK inhibitors suggest that the modification of the Adenosine kinase-mediated pathway has a potential role in cancer therapy [27,28,29]. GSK-3β plays a significant role in neurodegenerative diseases and diabetes, and has been investigated for the anticancer disease treatment strategies as well, where it has produced fruitful results [30,31,32]. Similarly, the MAPK enzyme is known for its significant role in the pathogenesis of cancer disease progression [33,34]. The designed compounds were found to have reasonably good interactions with all of the above proteins. It can be speculated that the observed cytotoxicity of the compounds might be attributed to the inhibition of one or more proteins involved with AK, GSK-3 β, and/or MAPK.

### 2.4. Potency of BTZ Derivatives to Inhibit GSK-3β

The compounds’ inhibitory activity was determined at the concentration of 50 µM, with SB-415286, a synthetic aryl indole derivative, serving as a positive reference. The inhibitory activity of all tested compounds is reported in Table 5, and was obtained from the average of experiments performed in triplicate. All of the tested compounds showed more than 60% inhibition at 50 µM concentration. Among the compounds, 2c, 2f, 2g, and 2j showed good inhibitory activity in these preliminary screening studies. It was found that the reference inhibitor compound, SB-415286, showed enzymatic inhibition of 98.62 ± 1.63% at the 20 µM concentration. Two compounds, 2c and 2j, were found to possess above 90% inhibitory action at the tested concentration range. 

Two compounds, 2f and 2j, showed sub-90% inhibitory effects, with activities of 89.70 ± 1.65 and 87.62 ± 2.54, respectively. The remaining compounds showed percentage inhibition in the range of 60.74 ± 2.85 to 76.35 ± 2.21 at 50 µM concentration. It was observed that the presence of a halogen atom at the 2′ and 4′ positions of the 4-phenyl pendant group might be crucial for activity, and it was expected that this halogen atom would play an important role in binding interactions with this enzyme’s active site. Similarly, the compounds containing the iodine substitution, such as 2a, 2b, 2d, and 2h, showed comparatively lower inhibitory potency, indicating that the bulky nature of the halogen may be involved in creating a steric crowding effect at the enzyme binding site. Most importantly, the OH group present on the 4-phenyl pendant was found to participate in critical H-bonding with the Asn195 of the cavity site, implying a positive effect on the inhibitory properties of the BTZ derivatives. It was further found that the incorporation of a hydrophobic group on the phenyl ring failed to improve the inhibitory properties of the BTZ compounds. The structural features of the BTZ nucleus might contribute to determining the best fitting in the binding cavity of the enzyme. The active conformations of the molecules for attaining the minimum energy levels were found to be distorted thiazepine rings. Extensive submicromolar-level activity studies may represent a detailed mechanistic approach to further establish these enzymes’ inhibition properties and the anticancer properties of the BTZ derivatives.

## 3. Materials and Methods

### 3.1. Synthesis of the 2,3-Dihydro-1,5-benzothiazepines

All reagents were procured from Sigma Aldrich India and were of synthetic grade. The progression of the reaction was monitored by TLC on pre-coated plates (silica gel 60F-254, 0.25 mm thick) from Merck, India, visualized in a UV Chamber and later with iodine vapors. A Shimadzu FT-IR spectrometer was used to record the IR spectra in KBr pellets. An Avance 300 MHz spectrometer was used to produce the ^1^H-NMR spectra in DMSO-d^6^, employing TMS as an internal standard. An EI-Shimadzu-GC-MS mass spectrometer was used to record the mass spectra of the synthesized compounds. A Carlo Erba 106 Perkin-Elmer model 240 analyzer was employed for elemental analysis.

#### The General Method Used for the Synthesis of 2,4-(Substituted-aryl)-2,3-dihydro-1,5-benzothiazepines

The derivatives of 1,5-benzothiazepine were synthesized as follows. Equimolar quantities of 2-Amino-4-methylbenzenethiol (1 mmol) and substituted 2′-hydroxy chalcone (1 mmol) were placed in a flat-bottomed flask, and a catalytic quantity of bleaching earth (10 wt.%, pH 12.5) in PEG-400 (20 mL) was added and stirred at 60–65 °C for 55 min (Figure 9). Product formation was confirmed by TLC (solvents: ethyl acetate and petroleum ether at 3:7). Later, the mixture was filtered in order to isolate the catalyst, the filtrate was poured into a beaker of ice-cold water (100 mL) while continuously stirring, and the resultant solid was vacuum filtered. The solid product was extracted, washed, and recrystallized with water (2 × 20 mL) and ethanol to produce the 1,5-benzothiazepine derivatives. Compound formation was confirmed using the Wilson test; none of the synthetic reactions showed the positive response signified by the red coloration of concentrated sulphuric acid [35,36,37].

*2-(2-(4-(dimethylamino)phenyl)-7-methyl-2,3-dihydrobenzo[b][1,4]thiazepin-4-yl)-4,6-diiodophenol (2a):* Yellow solid, Percentage yield: 88, m.p. 230–232 °C. IR (KBr):3169 (OH), 3042 (C-H arom), 1570 (C=N), 638 (C-Br); ^1^H NMR (300 MHz, DMSO- δ6):δ 10.42 (s, 1H, OH), δ 6.81–7.93 (m, 10H, ArH), 5.42 (dd, Hx, Jax = 10.81 Hz, Jbx = 2.12 Hz, 1H), 3.53 (dd, Hb, Jbx = 2.1 Hz, Jab = 14.1 Hz, 1H), 3.12 (dd, Ha, Jax = 10.82 Hz, Jab = 14.0 Hz, 1H), δ 2.1 (s, 3H, CH_3_), δ 2.79 (s, 6H, (CH_3_)_2_ ^13^C NMR (125 MHz, CDCl_3_, ppm) δ: 162.6, 159.1, 155.5, 149.5, 147.6, 136.6, 136.1, 133.0, 130.4, 128.6, 128.6, 128.0, 124.0, 122.0, 119.2, 112.9, 112.9, 88.6, 83.8, 50.1, 41.3, 41.3, 40.4,21.3; MS: m/z (%) 639 (M^+1^), Anal. Calcd. for C_24_H_22_I_2_N_2_OS: C,45.02; H, 3.47; N, 4.37; Found: C, 45.01; H, 3.43; N, 4.35%.

*2-(2-(4-(dimethylamino)phenyl)-7-methyl-2,3-dihydrobenzo[b][1,4]thiazepin-4-yl)-6-iodo-4-methylphenol (2b):* White solid, Percentage yield: 89, m.p. 218–220 °C. IR (KBr):3176(OH), 3052(C-H arom), 1582 (C=N); ^1^H NMR (300 MHz, DMSO- δ6):δ 10.39 (s, 1H, OH), δ 6.81–7.90 (m, 10H, ArH), 5.43 (dd, Hx, Jax = 10.79 Hz, Jbx = 2.10 Hz, 1H), 3.52 (dd, Hb, Jbx = 2.12 Hz, Jab = 14.1 Hz, 1H), 3.10 (dd, Ha, Jax = 10.81 Hz, Jab = 14.1 Hz, 1H), δ 2.72(s, 6H, (CH_3_)_2,_ δ 2.39 (s, 3H, CH_3_), δ 2.18 (s, 3H, CH_3_), ^13^C NMR (125 MHz, CDCl_3_, ppm) δ: 162.6, 157.2, 155.5, 149.5, 143.6, 136.1, 133.0, 132.7, 130.4, 129.7, 128.6, 128.6, 128.0, 124.0, 120.3, 119.2, 112.9, 112.9, 87.5, 50.1,41.3, 41.3, 40.4, 21.3,20.4 ; MS: m/z (%) 528 (M^+1^), Anal. Calcd for C_25_H_25_IN_2_OS: C,56.82; H, 4.77; N, 5.30; Found: C, 56.79; H, 4.75; N, 5.29%.

*2,4-dichloro-6-(2-(4-(dimethylamino)phenyl)-7-methyl-2,3-dihydrobenzo[b][1,4]thiazepin-4-yl)phenol (2c):* Yellow solid, Percentage yield: 92, m.p. 222–224 °C. IR (KBr):3187(OH), 3068(C-H arom), 1570(C=N), 732(C-Cl); ^1^H NMR (300 MHz, DMSO- δ6):δ 10.40 (s, 1H, OH), δ 6.81–7.93 (m, 10H, ArH), 5.39 (dd, Hx, Jax = 10.80 Hz, Jbx = 2.11 Hz, 1H), 3.50 (dd, Hb, Jbx = 2.14 Hz, Jab = 14.2 Hz, 1H), 3.14 (dd, Ha, Jax = 10.81 Hz, Jab = 14.2 Hz, 1H), δ 2.19 (s, 3H, CH_3_), δ 2.70(s, 6H, (CH_3_)_2_ ^13^C NMR (125 MHz, CDCl_3_, ppm) δ: 162.6, 157.9, 155.5, 149.5, 136.1, 134.1, 133.0, 130.4, 128.7, 128.6, 128.6, 128.4, 128, 126.9, 124.0, 121.6, 119.2, 112.9, 112.9, 50.1, 41.3, 41.3, 40.4, 21.3; MS: m/z (%) 456 (M^+1^), Anal. Calcd for C_24_H_22_Cl_2_N_2_OS: C,63.02; H, 4.85; N, 6.12; Found: C, 63.00; H, 4.82; N, 6.09%.

*4-chloro-2-(2-(4-(dimethylamino)phenyl)-7-methyl-2,3-dihydrobenzo[b][1,4]thiazepine-4-yl)-6-iodophenol (2d):* Creamy white solid, Percentage yield: 94, m.p. 244–246 °C. IR (KBr):3310(OH), 3056(C-H arom), 1587(C=N), 732(C-Cl); ^1^H NMR (300 MHz, DMSO- δ6):δ 10.48 (s, 1H, OH), δ 6.81–7.93 (m, 10H, ArH), 5.41 (dd, Hx, Jax = 10.81 Hz, Jbx = 2.11 Hz, 1H), 3.53 (dd, Hb, Jbx = 2.12 Hz, Jab = 14.2 Hz, 1H), 3.15 (dd, Ha, Jax = 10.81 Hz, Jab = 14.4 Hz, 1H), δ 2.14 (s, 3H, CH_3_), δ 2.72(s, 6H, (CH_3_)_2_ ^13^C NMR (125 MHz, CDCl_3_, ppm) δ: 162.6, 158.3, 155.5, 149.5, 141.5, 136.1, 133.0, 130.4, 129.5, 128.6, 128.6, 128.6, 128.0, 124.0, 121.8, 119.2, 112.9, 112, 89.0, 50.1, 41.3, 41.3, 40.4, 21.3; MS: m/z (%) 548 (M^+1^), Anal. Calcd for C_24_H_22_ClIN_2_OS: C,52.52; H, 4.05; N, 6.09; Found: C, 52.49; H, 4.02; N, 5.09%.

*2-bromo-6-(2-(4-(dimethylamino)phenyl)-7-methyl-2,3-dihydrobenzo[b][1,4]thiazepine-4-yl)-4-methylphenol (2e):* brown solid, Percentage yield: 86, m.p. 244–246 °C. IR (KBr):3174(OH), 3042(C-H arom), 1575(C=N), 629(C-Br); ^1^H NMR (300 MHz, DMSO- δ6):δ 10.43 (s, 1H, OH), δ 6.80–7.93 (m, 10H, ArH), 5.43 (dd, Hx, Jax = 10.82 Hz, Jbx = 2.1 Hz, 1H), 3.52 (dd, Hb, Jbx = 2.1 Hz, Jab = 14.2 Hz, 1H), 3.12 (dd, Ha, Jax = 10.82 Hz, Jab = 14.0 Hz, 1H), δ 2.72 (s, 6H, (CH_3_)_2,_ δ 2.36 (s, 3H, CH_3_), δ 2.1 (s, 3H, CH_3_), ^13^C NMR (125 MHz, CDCl_3_, ppm) δ: 162.6, 158.2, 155.5, 149.5, 137.5, 136.1, 133.3, 133.0, 130.4, 129.8, 128.6, 128.6, 128.0, 124.0, 120.9, 119.2, 115.1, 112.9, 112.9, 50.1, 41.3,41.3,40.4,21.3,20.6; MS: m/z(%) 480 (M^+1^), Anal. Calcd for C_25_H_25_BrN_2_OS: C,62.37; H, 5.23; N, 5.82; Found: C, 62.33; H, 5.20; N, 5.79%.

*2-bromo-4-chloro-6-(2-(4-(dimethylamino)phenyl)-7-methyl-2,3-dihydrobenzo [b][1,4]thiaz epin-4-yl)phenol (2f):* Yellow solid, Percentage yield: 87, m.p. 244–246 °C. IR (KBr):3171(OH), 3039(C-H arom), 1572(C=N), 632(C-Br); ^1^H NMR (300 MHz, DMSO- δ6):δ 10.45 (s, 1H, OH), δ 6.81–7.93 (m, 10H, ArH), 5.45 (dd, Hx, Jax = 10.82 Hz, Jbx = 2.1 Hz, 1H), 3.54 (dd, Hb, Jbx = 2.1 Hz, Jab = 14.2 Hz, 1H), 3.10 (dd, Ha, Jax = 10.82 Hz, Jab = 14.0 Hz, 1H), δ 2.2 (s, 3H, CH_3_), δ 2.70 (s, 6H, (CH_3_)_2_ ^13^C NMR (125 MHz, CDCl_3_, ppm) δ: 162.6, 159.3, 155, 149, 136.2, 136.1, 133.0, 130, 129.6, 129, 128.6, 128.6,128.0, 124,122.4, 119.2, 115.3, 112.9, 112.9, 50.1, 41.3,41, 40.4, 21.3; MS: m/z (%) 500 (M^+1^), Anal. Calcd for C_24_H_22_BrClN_2_OS: C,57.44; H, 4.42; N, 5.58; Found: C, 57.43; H, 4.40; N, 5.56%.

*2,4-dibromo-6-(2-(4-(dimethylamino)phenyl)-7-methyl-2,3-dihydrobenzo[b][1,4]hiazepine-4-yl)phenol (2g):* Light brown solid, Percentage yield: 89, m.p. 244–246 °C. IR (KBr):3179(OH), 3056(C-H arom), 1579(C=N), 645(C-Br); ^1^H NMR (300 MHz, DMSO- δ6):δ 10.39 (s, 1H, OH), δ 6.80–7.92 (m, 10H, ArH), 5.39 (dd, Hx, Jax = 10.80 Hz, Jbx = 2.12 Hz, 1H), 3.45 (dd, Hb, Jbx = 2.2 Hz, Jab = 14.2 Hz, 1H), 3.09 (dd, Ha, Jax = 10.82 Hz, Jab = 14.1 Hz, 1H), δ 2.1 (s, 3H, CH_3_), δ 2.69 (s, 6H, (CH_3_)_2_ ^13^C NMR (125 MHz, CDCl_3_, ppm) δ: 162.6, 160.2, 155.5, 149.5, 138.4, 136.1, 133.1, 133.0, 130.4,128.6,128.6,128.0,124.0 123.2, 119.2, 116.1, 113.5, 112.9, 112.9, 50.1, 41.3, 41.3, 40.4, 21.3; MS: m/z (%) 545 (M^+1^), Anal. Calcd for C_24_H_22_Br_2_N_2_OS: C,52.76; H, 4.06; N, 5.13; Found: C, 52.72; H, 4.05; N, 5.10%.

*4-bromo-2-(2-(4-(dimethylamino)phenyl)-7-methyl-2,3-dihydrobenzo[b][1,4]thiazepin-4-yl)-6-iodophenol (2h):* brown solid, Percentage yield: 85, m.p. 244–246 °C. IR (KBr):3197(OH), 3084(C-H arom), 1569(C=N), 652(C-Br); ^1^H NMR (300 MHz, DMSO- δ6):δ 10.42 (s, 1H, OH), δ 6.81–7.93 (m, 10H, ArH), 5.38 (dd, Hx, Jax = 10.81 Hz, Jbx = 2.11 Hz, 1H), 3.43 (dd, Hb, Jbx = 2.1 Hz, Jab = 14.2 Hz, 1H), 3.12 (dd, Ha, Jax = 10.82 Hz, Jab = 14.2 Hz, 1H), δ 2.2 (s, 3H, CH_3_), δ 2.70 (s, 6H, (CH_3_)_2_ ^13^C NMR (125 MHz, CDCl_3_, ppm) δ: 162.6, 159.2, 155.5, 149.5, 144.4, 136.1, 133.0, 133.0, 130.4, 128.6, 128.6, 128.0,124.0,122.6, 119.2, 117.4, 112.9, 112.9, 89.8, 50.1, 41.3, 41.3, 40.4, 21.3; MS: m/z (%) 591 (M^+1^), Anal. Calcd for C_24_H_22_BrIN_2_OS: C,48.58; H, 3.74; N, 4.72; Found: C, 48.55; H, 3.72; N, 4.69%.

### 3.2. Biological Evaluation

#### Anti-Proliferative Activity

Cell culture:

HepG-2, and DU-145 human tumor cell lines were grown in DMEM medium supplemented with Glutamax-I and glucose. A pseudo-normal human embryonic liver cell line (L02) was used to study the cytotoxicity of the synthesized compounds. The cell culture medium was supplemented with fetal calf serum (10% *v*/*v*), penicillin (100 IU/mL), and streptomycin (100 µg/mL), then preserved at 37 °C. Prior to cytotoxicity evaluation, cell viability tests were executed employing the trypan blue exclusion method and allowed to progress further if the cells showed more than 95% of viability.

Cytotoxicity assay:

A spectrophotometric MTT reagent assay was employed to assess the cytotoxic activity of the synthesized compounds. In brief, the grown cells (approx. 5 × 10^3^ count) were placed in each well of a 96-well plate and 100 µL of the culture medium was added along with the synthesized compounds (1–100 µmol/L). The well plate was subjected to incubation for 72 h; later, 10 µL of MTT (5 mg/mL) was added to each well. The well plate was further incubated for 4 h at 37 °C. The resulting insoluble formazan was dissolved by adding sodium dodecyl sulfate (100 µL, 10%) to each well and continuing incubation for an additional 12 h. Later, the 96-well plate was used to measure the absorbance at the 540 nm and 630 nm reference wavelengths using the spectrophotometer. The blank absorbance was treated as a control and the IC_50_ values were determined by a non-linear regression model with the help of normalized dose-response data acquired via MTT assay [38,39].

### 3.3. Molecular Docking

Molecular modeling studies were performed using AutoDock 4.2.6 (The Scripps Research Institute, CA 92037, USA) installed on a Lenovo PC with a Core i3 processor and a Windows 10 operating system. The structures were drawn in ChemDraw 18.2 software, and the Protein–Ligand interactions were visualized in the Biovia Discovery Studio 2020 Client program.

#### 3.3.1. Small Molecule Preparation

The chemical structures of the BTZ derivatives considered in the docking study were drawn using ChemDraw 18.2 software (PerkinElmer Informatics, Waltham, Massachusetts, USA). All of the structures were cleaned and optimized in Chem3D 18.2. The energy optimization of the molecules was performed in Chem3D 18.2 using Merck Molecular Force Field (MMFF) with a distance-dependent dielectric function and an energy gradient of 0.001 kcal/mol Å.

#### 3.3.2. Protein Preparation

BTZ compounds were reported to possess kinase enzyme inhibitory properties. The kinase proteins Human Adenosine Kinase (PDB ID: 2I6B), Glycogen synthase kinase-3β (GSK-3 β, PDB ID: 1Q41), and Human Mitogen-Activated Protein Kinase 1 (MEK1, PDB ID: 3w8q) were selected for docking studies (Table 6).

The 3D structures of the proteins were retrieved from the RCSB PDB database as complexes bound with their respective co-crystallized inhibitors (Table 6). The ligand and water molecules were removed from the protein and later assigned with polar hydrogens and Kollman charges. The designed molecules were minimized and the docking analysis was performed on the prepared proteins. The ligand-binding interactions with the respective proteins were compared with the co-crystalized ligands using a standard docked method, the same one used for the calculation of the RMSD of the docked molecules.

#### 3.3.3. Docking Methodology

2I6B: The AutoGrid was generated by setting the grid map with an input (number of points in XYZ) of 46–40–40 Å as the grid box to encircle the original ligand. The grid box space was 0.403 Å length and was centered with the following dimensions: x = 5.603, y = −1.159 and z = 25.349. Using the co-crystallized ligand structure (89I), the scoring grid was designed to reduce the computation time. The AutoDock program was performed with default retries and generations of 10,000 and 27,000, respectively. The Lamarckian genetic algorithm was chosen for the conformational search of ligands. AutoGrid 4 was employed to define the binding area to dock the molecules. While defining the grid, the following atomic types were set: for macromolecules, A (aromatic Carbon), C, HD, N, OA, SA, NA; for ligands, A, C, Cl, NA, OA, N, SA, HD. Additional docking parameters were kept at their default settings, with 2Å RMSD for clustering the conformations. 

1Q41: For the GSK-3β protein, the grid box was set with a space of 0.476 Å length and centered with the following dimensions: x = 8.329, y = −3.265, and z = 18.781.

3w8q: For the MEK1 protein, the grid box was set with a space of 0.492 Å length and centered with the following dimensions: x = 6.523, y = −2.546, and z = 21.148.

### 3.4. GSK-3β Inhibitory Assay

The GSK-3β inhibition assay was performed as per the method described by Baki et al. [43,44], with a Kinase-Glo^®^ Max Kit (Promega Biotech India Pvt. Ltd., New Delhi, India). GSK-3β (His-Tag, Human, Recombinant) and GSM substrate were obtained from Sigma-Aldrich, Bangalore, India. The remaining materials used in the luminescence assay, such as ATP disodium salt hydrate, ammonium acetate, ammonium hydroxide, 4-(2- hydroxyethyl)piperazine-1-ethanesulfonic acid (HEPES), ethylene glycol- bis(-aminoethylether)-N, N, N, N-tetra-acetic acid tetrasodium salt (EGTA), ethylenediaminetetraacetic acid (EDTA), dimethyl sulfoxide (DMSO), magnesium acetate tetrahydrate, formic acid, 3-[(3-chloro-4-hydroxy phenyl)amino]-4-(2-nitrophenyl)-1H-pyrrol-2,5-dione, and GSK-3β selective inhibitor SB-415286, were all procured from Sigma-Aldrich, Bangalore, India. The luminescence assay was carried out using an 800 TS Absorbance Reader (Agilent Technologies Pvt. Ltd. Hyderabad, India). 

Briefly, the assay procedure was as follows. Each microplate well contained 10 µL of the test compound (1mM, in DMSO) in a pH 7.5 buffer comprising 50 mM HEPES, 1 mM each of EGTA, EDTA, and 15 mM magnesium acetate, which contained ATP (1 M, 10 µL), GSM (10 µL, 100 µM), and GSK-3β (20 ng, 10 µL). In order to obtain the positive and negative controls, ten microliters of either buffer or SB-415286 solution (5 µM final concentration) were added instead of the test compound solution. The final DMSO content in the reaction mixture did not surpass 5%. The mixture was allowed to react for 30 min at 37 °C, and the enzymatic processes were then halted using 40 µL of Kinase-Glo reagent and incubated for 10 min. The activity was determined by the difference between total and consumed ATP. The inhibition activity was determined using the maximal activity in the absence of the inhibitor and the maximal inhibition in the presence of the reference drug. IC50 values were determined using the GraphPad Prism 4.0 tool (GraphPad Software Inc., San Diego, CA, USA).

## 4. Conclusions

In this research, we have described a cost-effective and practically workable process protocol for the development of a new series of bioactive 1,5-benzothiazepine derivatives. Application of the bleach tone basic medium catalysis method and the use of a green PEG-400 solvent were found to be efficient in achieving the synthesis of the titled compounds in good yields. The in vitro cytotoxic activity of the synthesized 1,5-benzothiazepine derivatives was investigated and found to be particularly active against liver cancer cell lines, with a single-digit micromolar range. Additionally, in silico studies were carried out to predict possible drug targets for the designed compounds. It was found that compounds containing the halogenated phenyl group substitution provided good anticancer activity, and this structural feature was corroborated by our molecular modelling simulations against three studied molecular targets. Further structural exploration of the 1,5-benzothiazepine core ought to be undertaken in order to develop more potent compounds as novel potential anticancer agents. 

## Figures and Tables

**Figure 1 molecules-27-03757-f001:**
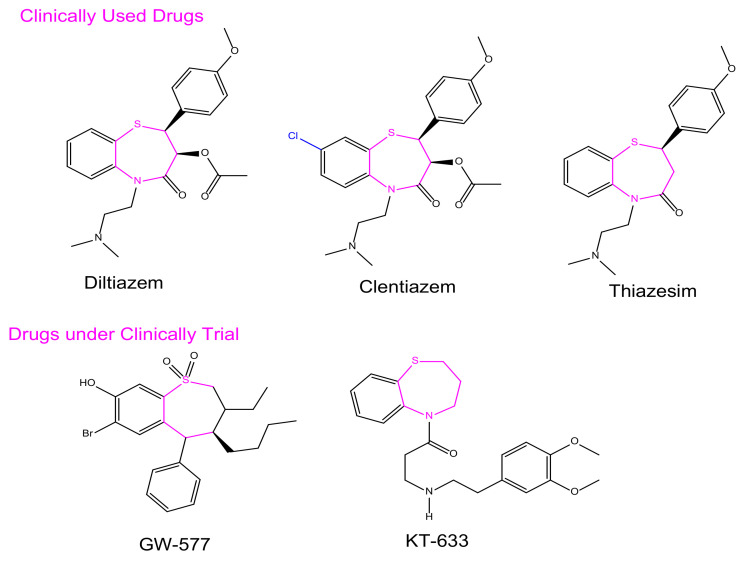
Representative BTZ derivatives either in clinical use or currently undergoing clinical trials.

**Figure 2 molecules-27-03757-f002:**
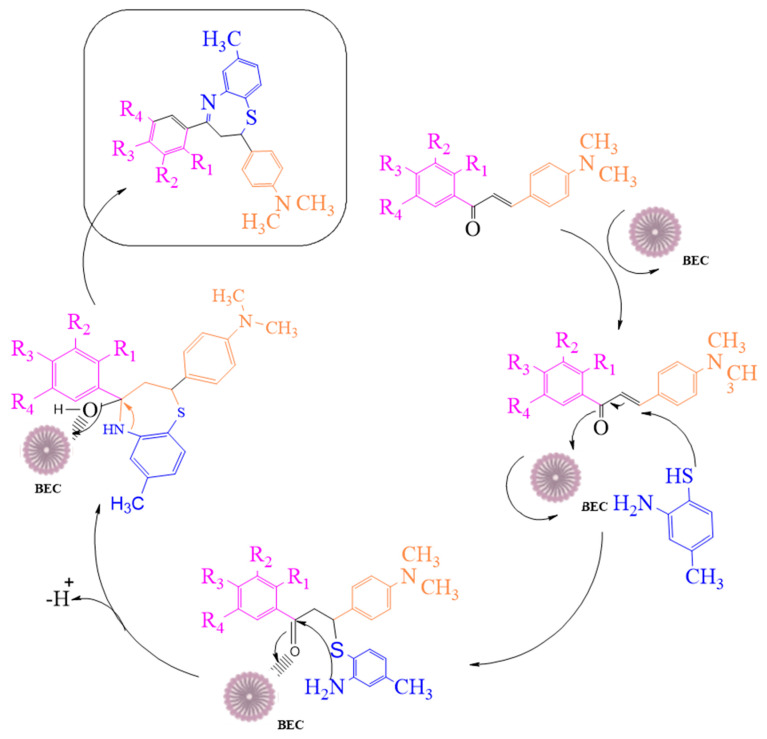
The mechanistic approach for the synthesis of 1,5-benzothiazepine derivatives.

**Figure 3 molecules-27-03757-f003:**
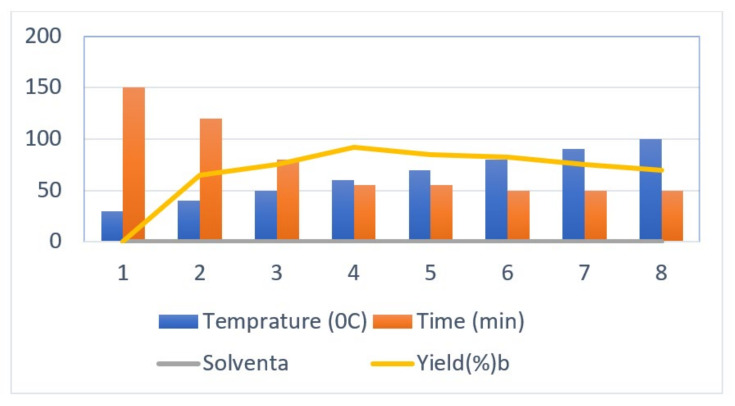
Optimization of temperature for the synthesis of 1,5-benzothiazepine derivatives.

**Figure 4 molecules-27-03757-f004:**
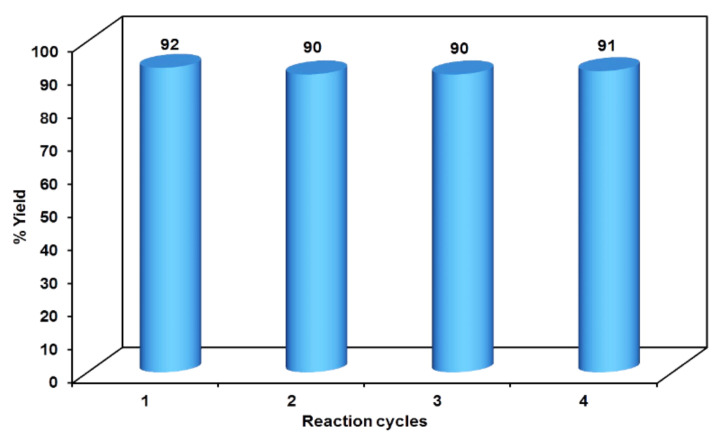
Reusability potency of PEG-400.

**Figure 5 molecules-27-03757-f005:**
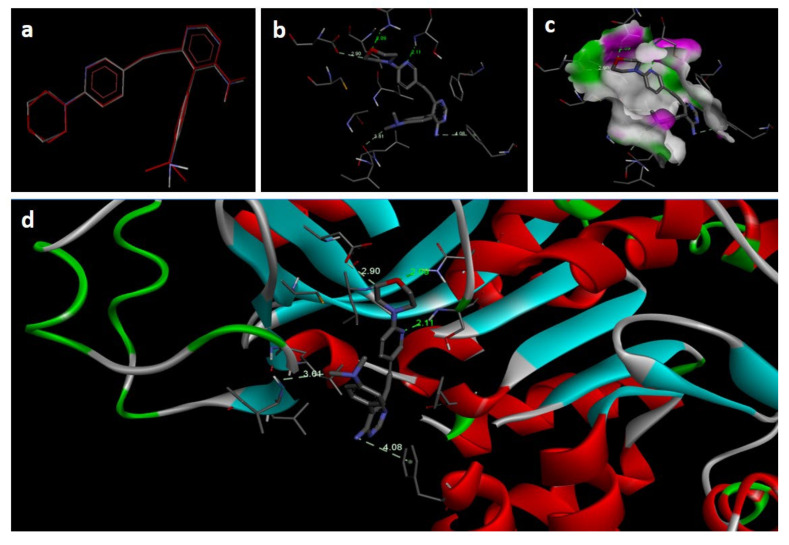
Docking images of co-crystallized ligand on 2I6B: (**a**) validation of docking method; (**b**) H-bonds between ligand (89I) and protein residues; (**c**,**d**) 3D representation of interactions in binding cavity.

**Figure 6 molecules-27-03757-f006:**
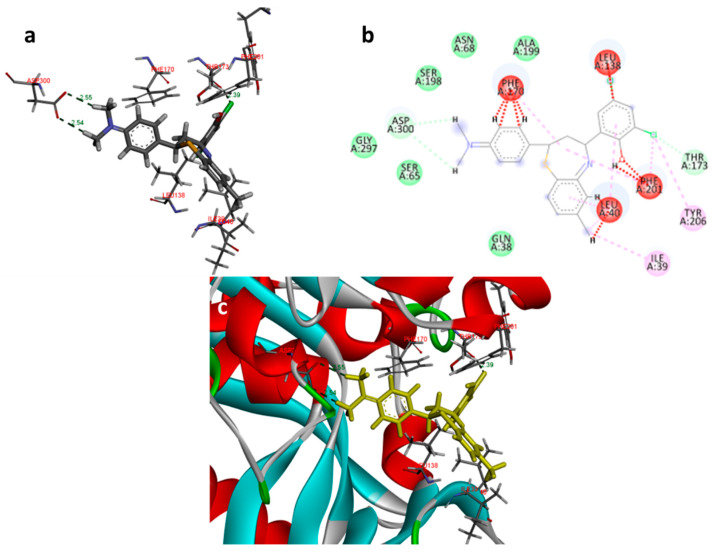
Docking images of compound 2c on 2I6B: (**a**) H-bonds between 2c and protein residues; (**b**) 2d representation of interactions in binding cavity; (**c**) 3D representation of interactions in the binding cavity.

**Figure 7 molecules-27-03757-f007:**
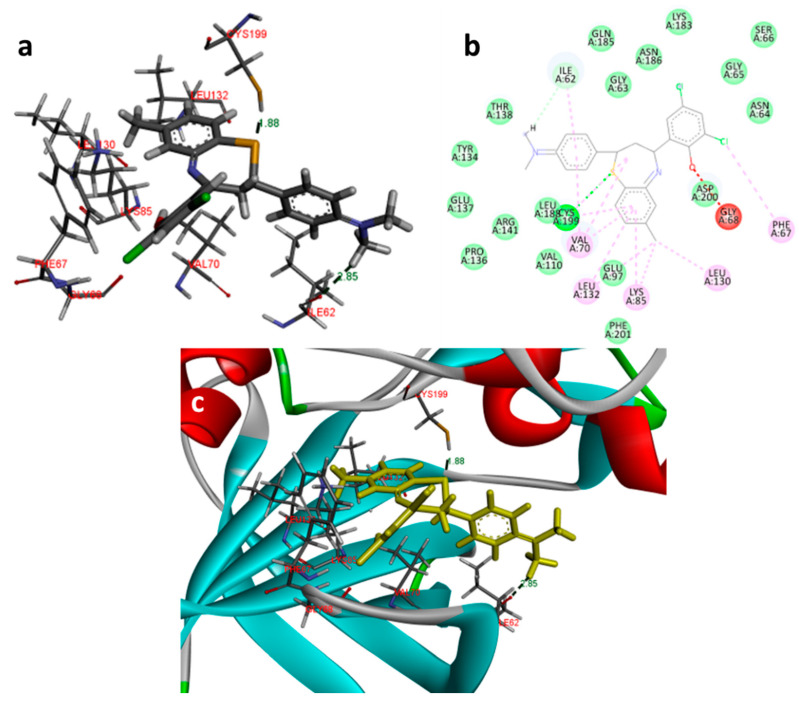
Docking images of 2f with 1Q41: (**a**) 3d representation of interactions; (**b**) 2D representation of interactions; (**c**) 3d orientation in the binding cavity of the protein.

**Figure 8 molecules-27-03757-f008:**
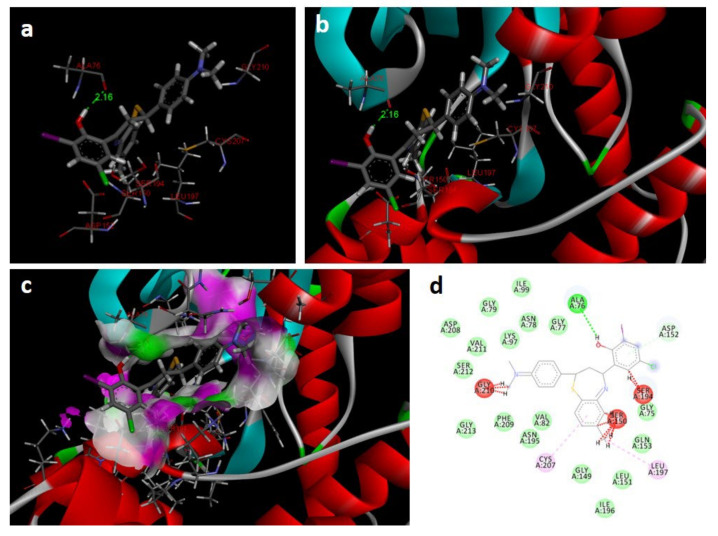
Docking images of 2c with 3w8q: (**a**) H-bonds between 2c and protein residues; (**b**,**c**) 3d representation of interactions in the binding cavity; (**d**) 2D representation of interactions in the binding cavity.

**Figure 9 molecules-27-03757-f009:**
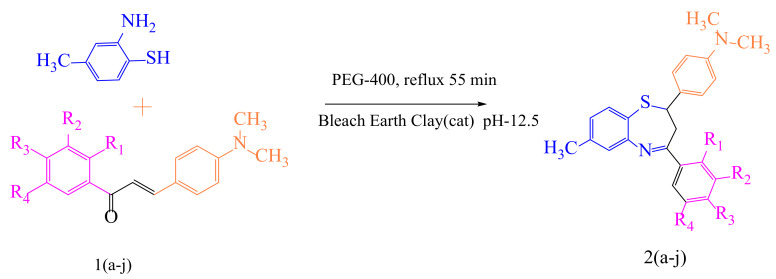
Scheme of 1,5-benzothiazepine derivatives in the catalytic mechanisms of PEG-400 mediated synthesis.

**Table 1 molecules-27-03757-t001:** Optimization of temperature for the synthesis of 1,5-benzothiazepine derivatives.

Entry No.	Temperatures (°C)	Period (min)	Solvent ^a^	Yield (%) ^b^
1	30	150	PEG-400	--
2	40	120	PEG-400	35
3	50	80	PEG-400	68
**4**	**60**	**55**	**PEG-400**	**92**
5	70	55	PEG-400	85
6	80	50	PEG-400	82
7	90	50	PEG-400	75
8	100	50	PEG-400	70

^a^ Reaction conditions: 1 (1 mM), 2 (1 mMl) at 60 (°C) and PEG-400 at 60 (°C), ^b^ Yield of isolated product.

**Table 2 molecules-27-03757-t002:** Effects of solvent on outcome of the reaction of 1,5-benzothiazepine derivatives at 60 °C.

Entry	Solvent	Period (min)	Yield (%)
1	EtOH	255	62
2	DCM	220	68
3	CH_3_CN	175	70
4	PEG-400	55	92

**Table 3 molecules-27-03757-t003:** Anticancer activity of compounds 2a–j in selected experimental human cancer cell lines.

Compound	Substitutions				IC_50_ (µM)		% Inhibition (100 µM)
	R_1_	R_2_	R_3_	R_4_	Hep G-2	DU-145	L02
2a	OH	I	H	I	7.52 ± 0.13	32.74 ± 0.13	28.23 ± 1.68
2b	OH	I	H	CH_3_	8.26 ± 0.26	41.34 ± 0.12	19.65 ± 0.97
**2c**	OH	Cl	H	Cl	3.29 ± 0.15	20.45 ± 0.19	19.29 ± 1.25
2d	OH	I	H	Cl	7.89 ± 0.22	36.17 ± 0.14	31.42 ± 1.14
2e	OH	Br	H	CH_3_	6.87 ± 0.15	37.52 ± 0.27	52.16 ± 1.42
**2f**	OH	Br	H	Cl	4.38 ± 0.11	24.58 ± 0.13	28.41 ± 1.18
2g	OH	Br	H	Br	8.32 ± 0.16	28.53 ± 0.14	15.56 ± 1.29
2h	OH	I	H	Br	8.56 ± 0.14	36.80 ± 0.18	46.12 ± 1.05
2i	OH	H	CH_3_	Cl	7.74 ± 0.08	40.64 ± 0.09	27.47 ± 1.32
**2j**	OH	H	H	Br	4.77 ± 0.21	15.42 ± 0.16	21.65 ± 1.03
Methotrexate	-				4.68 ± 0.17	21.96 ± 0.15	78.23 ± 1.86

**Table 4 molecules-27-03757-t004:** Docking results against adenosine kinase, glycogen synthase kinase-3β, and mitogen-activated protein kinase 1.

S. No	Ligand No.	AK		GSK-3 β		MAPK	
		Bond Energy	K_i_ (µM)	Bond Energy	K_i_ (µM)	Bond Energy	K_i_ (µM)
1	Native ligand	−7.88	1.66	−8.01	9.03	−14.78	21.23
2	2a	−5.22	11.87	−5.05	12.52	−2.37	69.17
3	2b	−6.39	7.36	−1.04	36.05	−0.35	76.97
4	2c	−7.65	2.49	−4.32	17.13	−6.82	51.98
5	2d	−7.41	3.42	−4.78	13.58	−3.97	62.99
6	2e	−2.68	21.68	−0.54	29.93	−3.26	65.73
7	2f	−4.58	14.35	−5.27	11.67	−2.86	67.27
8	2g	−5.36	11.33	−0.04	32.20	−0.95	74.65
9	2h	−4.85	13.30	−0.74	29.18	−0.40	76.78
10	2i	−4.08	16.28	−1.78	25.14	−4.06	62.64
11	2j	−3.99	16.63	−1.76	25.25	−11.71	33.09

**Table 5 molecules-27-03757-t005:** Screening of the title compounds for their potency to inhibit GSK-3β (50 µM).

Compound	R1	R2	R3	R4	% of Inhibition at 50 µM *
2a	OH	I	H	I	62.35 ± 2.21
2b	OH	I	H	CH_3_	60.74 ± 2.85
2c	OH	Cl	H	Cl	95.21 ± 3.64
2d	OH	I	H	Cl	61.14 ± 1.98
2e	OH	Br	H	CH_3_	66.63 ± 2.51
2f	OH	Br	H	Cl	89.70 ± 1.65
2g	OH	Br	H	Br	87.62 ± 2.54
2h	OH	I	H	Br	61.93 ± 3.47
2i	OH	H	CH_3_	Cl	61.52 ± 3.26
2j	OH	H	H	Br	92.65 ± 1.89
		SB-415286			98.62 ± 1.63

* Except SB-415286, which was tested at a concentration of 20 µM.

**Table 6 molecules-27-03757-t006:** Protein acquisition data.

PDB ID	Co-Crystallized Ligand or Inhibitor	Resolution	Reference
2I6B	5-[4-(Dimethylamino)Phenyl]-6-[(6-Morpholin-4-yl pyridin-3-yl)Ethynyl]Pyrimidin-4-Amine (89I)	2.30 Å	[40]
1Q41	(Z)-1h,1’h-[2,3’]Biindolylidene-3,2’-Dione-3-Oxime (IXM)	2.10 Å	[41]
3w8q	Phosphothiophosphoric Acid-Adenylate Ester (AGS)	2.20 Å	[42]

## Data Availability

Not applicable.

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
