# Peer review of "1,5-Benzothiazepine Derivatives: Green Synthesis, In Silico and In Vitro Evaluation as Anticancer Agents"

_molecules, 2022, doi:10.3390/molecules27123757_

Round 1

Reviewer 1 Report

The manuscript molecules-1740233 devoted the actual field of medicinal chemistry, namely design and green synthesis of 1,5-benzothiazepine as antitumor agents and can be interested to the specialists working in this field. The authors’ opinion is clear and based on a good experimental material. The paper fit the Journal scope and formal requirements. However, it needs major revision before publication.

To improve the quality and perception of the manuscript I would suggest paying attention to following comments:

  1. Abstract in general should be revised. The abstract should be more focused on the topic of the article. It should highlight the main points of the experimental data. The first two sentences are general, so they need to be removed.
  2. The experimental part should be presented as it is generally accepted in articles of a chemical profile. Therefore, spectral and physicochemical data should be transferred from Supplementary to the main text of the manuscript. Table 3 must be removed and its data transferred to the experimental part.
  3. The assignment of the 13C NMR data would be an added value.
  4. The manuscript lacks the clear biological focus. SAR should be discussed more deeply; probably the explanation of the effect of certain structural fragments on activity would be good to discuss.
  5. The authors claimed anticancer activity. But the reviewer doubts that these data is good enough for the development of novel drug-like molecules. In order to understand the nature of the antitumor effect of the synthesized derivatives the cytotoxicity toward pseudonormal cells should be studied. Authors should substantially review the biological part of the work.
  6. There are some grammar and orthographical errors in the manuscript, which should be corrected

My decision is major revision.

Author Response

Manuscript: molecules-1740233

Thank you for reviewing our manuscript. We have incorporated the suggested changes in the manuscript. Here are our specific responses to the reviewers’ comments. 

Thank you for your kind consideration.

Regards

Point 1: Abstract in general should be revised. The abstract should be more focused on the topic of the article. It should highlight the main points of the experimental data. The first two sentences are general, so they need to be removed.

Response: As per the suggestions, the general text was removed from the abstract and the experimental and results related data has been incorporated.

Point 2: The experimental part should be presented as it is generally accepted in articles of a chemical profile. Therefore, spectral and physicochemical data should be transferred from Supplementary to the main text of the manuscript. Table 3 must be removed and its data transferred to the experimental part.

Response: The physicochemical and spectral characterization data of the synthesized compounds have been mentioned in the main text of the manuscript under the methodology section as the suggestions.

Point 3: The assignment of the 13C NMR data would be an added value.

Response: In the original supplementary material that was uploaded earlier has the 13C NMR data for almost all the synthesized compounds, now as per the suggestions, we have moved that into the methodology part of the manuscript.

Point 4: The manuscript lacks a clear biological focus. SAR should be discussed more deeply; probably the explanation of the effect of certain structural fragments on activity would be good to discuss.

Response: As per the suggestions, we have mentioned the in-depth discussion about the structural features and the corresponding influence on biological activity.

Point 5: The authors claimed anticancer activity. But the reviewer doubts that these data is good enough for the development of novel drug-like molecules. In order to understand the nature of the antitumor effect of the synthesized derivatives the cytotoxicity toward pseudonormal cells should be studied. Authors should substantially review the biological part of the work.

Response: We apologize for the missing data, initially we have included some antimicrobial activity data also in the manuscript and later was removed for various reasons, unfortunately, the cytotoxicity data was also lost in the process. Now it has been updated in the manuscript. Thank you for the suggestion.

Point 6: There are some grammar and orthographical errors in the manuscript, which should be corrected

Response: The grammatical errors and the scientific expressions were corrected by a fluent native English speaker.

Reviewer 2 Report

Paper titled 1, 5-Benzothiazepine derivatives: Green Synthesis, In Silico and 2
In Vitro Evaluation as Anticancer Agents describes the synthesis, fundamental spectroscopic analysis and evaluation of the obtained compounds as antibacterial agents.

Abstract needs revision, it should be a synthetic description of the entire thesis. The fragment: Benzothiazepines have immense importance as a heterocyclic moiety and found in a va- 34
riety of bioactive compounds. It can be synthesized by various synthetic routes, act as a main phar- 35 macophore to achieve the bioactivity and serving as significant template in synthetic and medicinal 36 chemistry. has to be removed.

English needs corrections.

chapter 2.1. Synthesis of the 2, 3-dihydro-1, 5-benzothiazepines

fragment : In the past two decades, the exploration of the alternative catalysts and solvents for organic synthesis has resulted with handful of materials with improved results in selectivity, yield, and safety. PEG-400 is one among them, which is considered as green chemical due to its nature, viz, eco-friendly, non-toxic, recyclability, mild reaction conditions and most importantly inexpensive [23-26] and has attracting the organic chemists. We 95 have attempted to describe a mild and efficient process for the synthesis of BTZ derivatives in the presence of PEG-400 and Bleaching earth clay support (Figure 2). should be removed. Authors can put these information in different chapter.

chapter 2.2.1. Anti-proliferative activity: should be rewritten.

The text in chapter: 2.3. Molecular Docking should be rewriten, The tested compounds shown the bond energy between 7.65 and -2.68 kcal/mol, which confirms the potential binding affinity of these compounds into the binding cavity of the enzyme. The docking method was validated by performing docking simulation of the drawn structure of the co-crystallized ligand (89I) and compared with the co-crystallized conformation present in the protein" was repeated twice.

Author Response

Dear Editor and Reviewer

Manuscript: molecules-1740233

Thank you for reviewing our manuscript. We have incorporated the suggested changes in the manuscript. Here are our specific responses to the reviewers’ comments (Typed in BLUE font)

Thank you for your kind consideration.

Regards

Reviewer 2

Comments and Suggestions for Authors

Paper titled 1, 5-Benzothiazepine derivatives: Green Synthesis, In Silico and In Vitro Evaluation as Anticancer Agents describes the synthesis, fundamental spectroscopic analysis and evaluation of the obtained compounds as antibacterial agents.

Point 1: Abstract needs revision, it should be a synthetic description of the entire thesis. The fragment: Benzothiazepines have immense importance as a heterocyclic moiety and found in a va- 34 riety of bioactive compounds. It can be synthesized by various synthetic routes, act as a main phar- 35 macophore to achieve the bioactivity and serving as significant template in synthetic and medicinal 36 chemistry. has to be removed.

Response: As suggested, general text from the abstract is removed. Result-related data has been incorporated.

Point 2: English needs corrections.

Response: Thank you for the advice. We got our manuscript corrected by a native English speaker expert in the field of synthetic chemistry.

Point 3: chapter 2.1. Synthesis of the 2, 3-dihydro-1, 5-benzothiazepines

fragment: In the past two decades, the exploration of the alternative catalysts and solvents for organic synthesis has resulted with handful of materials with improved results in selectivity, yield, and safety. PEG-400 is one among them, which is considered as green chemical due to its nature, viz, eco-friendly, non-toxic, recyclability, mild reaction conditions and most importantly inexpensive [23-26] and has attracting the organic chemists. We 95 have attempted to describe a mild and efficient process for the synthesis of BTZ derivatives in the presence of PEG-400 and Bleaching earth clay support (Figure 2). should be removed. Authors can put these information in different chapter.

Response: We understood your suggestion and change has been made accordingly.

Point 4: chapter 2.2.1. Anti-proliferative activity: should be rewritten.

The text in chapter: 2.3. Molecular Docking should be rewriten, The tested compounds shown the bond energy between 7.65 and -2.68 kcal/mol, which confirms the potential binding affinity of these compounds into the binding cavity of the enzyme. The docking method was validated by performing docking simulation of the drawn structure of the co-crystallized ligand (89I) and compared with the co-crystallized conformation present in the protein" was repeated twice.

Response: Suggestion implemented. 

Reviewer 3 Report

Paper (molecules-1740233) entitled “1, 5-Benzothiazepine derivatives: Green Synthesis, In Silico and In Vitro Evaluation as Anticancer Agents” by Islam………………………. Tratrat reports. The paper must be rewritten carefully because there are various flaws in general and even in the scientific approach. I am not able to recommend this paper in its present form.

In the abstract section, include a summary of results of your findings like yields, IC50, and binding score/energy etc. in order to get an idea without reading the entire paper.

In Introduction section, some paragraphs are too complicated to understand, such as " Numerous efforts …………. shown special interest”, it is better to split these types of paragraphs.

In some paragraphs, grammar and typographical errors should be corrected.

Figure 2, gives the same color of particular compounds in the entire reaction mechanism cycle in a tentative scheme (mentioned by blue color and next step by red color etc.).

Mention explicitly about the reactants, either 2-Amino-4-methylbenzenethol or 2-Aminothiophenol was used in the reaction mechanism (Fig 2) as well as in the synthesis section.

What is the Mechanical approach in “Figure 2. Mechanical approach of 1,5 derivatives of benzothiazepine.) or you want to say “mechanistic approach”

Grammar of the paper should be polished by a native speaker (……has attracting the organic chemists or …. has attracted…….)

In the section Results and Discussion, under heading of “Synthesis of the 2, 3-dihydro-1, 5-benzothiazepines” Significance of bleaching earth is not explained, why; is it irrelevant to the reaction's execution?

Provide spectral data (IR, NMR, etc.) for any of the compounds as supplementary information

In the biological section, antimicrobial was not found in the manuscript as mentioned in the introduction lines (75-78) (As part of our ongoing efforts to identify new chemical entities (NCEs) endowed with biological activity, we have considered the possibility of using a novel combination approach to the BTZ scaffold to investigate its antimicrobial and anticancer properties)

Figure 6-7 shows that docking interactions are not perfect because some amino acids, which are part of the active site, present obstacles and lead to destabilization of the protein-ligand complex.

The authors did not mention the nature of bleaching earth, whether it was used as is or in activated form. Did you regenerate after isolation of the catalyst (bleaching earth) for further use in the reaction?

Check this sentence logically, “catalytic quantity of bleaching earth (10 moles, pH 12.5) (line; 304)” Do you have any idea when a catalytic amount/quantity is written or called?

Author Response

Dear Editor and Reviewer

Manuscript: molecules-1740233

Thank you for reviewing our manuscript. We have incorporated the suggested changes in the manuscript. Here are our specific responses to the reviewers’ comments.

Thank you for your kind consideration.

Regards

Point 1: In the abstract section, include a summary of results of your findings like yields, IC50, and binding score/energy etc. in order to get an idea without reading the entire paper.

Response: As per the suggestions, the general text was removed from the abstract and the experimental and results related data has been incorporated.

Point 2: In Introduction section, some paragraphs are too complicated to understand, such as " Numerous efforts …………. shown special interest”, it is better to split these types of paragraphs.

Response: Agreed. Suggestion implemented. The highlighted paragraph on green chemical synthesis importance and followed by BTZ synthesis approaches was shifted to the suitable place for continuity. Corresponding references are aligned accordingly.

Point 3: In some paragraphs, grammar and typographical errors should be corrected.

Response: Thank you for the advice. We got our manuscript corrected by a native English speaker expert in the field of synthetic chemistry.

Point 4: Figure 2, gives the same color of particular compounds in the entire reaction mechanism cycle in a tentative scheme (mentioned by blue color and next step by red color etc.).

Response:  In the revised manuscript, we have followed uniform color representation throughout the scheme based on the source of rings from starting materials.

Point 5: Mention explicitly about the reactants, either 2-Amino-4-methylbenzenethiol or 2-Aminothiophenol was used in the reaction mechanism (Fig 2) as well as in the synthesis section.

Response: We have changed the 2-Aminothiophenol with 2-Amino-4-methylbenzenethiol reactant in the manuscript.

Point 6: What is the Mechanical approach in “Figure 2. Mechanical approach of 1,5 derivatives of benzothiazepine.) or you want to say “mechanistic approach”

Response: The intension is to represent the probable mechanism of reaction, so modified as Mechanistic approach as per the reviewer suggestions.

Point 7: Grammar of the paper should be polished by a native speaker (……has attracting the organic chemists or …. has attracted…….)

Thank you for the advice. We got our manuscript corrected by a native English speaker expert in the field of synthetic chemistry.

Point 8: In the section Results and Discussion, under heading of “Synthesis of the 2, 3-dihydro-1, 5-benzothiazepines” Significance of bleaching earth is not explained, why; is it irrelevant to the reaction's execution?

Response: Bleaching earth clay is one of the good heterogeneous catalysts and has a significant role in the current synthesis. Its significance and role in the current study have been updated in the main text of the manuscript.

Point 9: Provide spectral data (IR, NMR, etc.) for any of the compounds as supplementary information.

Response: We have included all the characterization details in the main manuscript under the Methods section.

Point 10: In the biological section, antimicrobial was not found in the manuscript as mentioned in the introduction lines (75-78) (As part of our ongoing efforts to identify new chemical entities (NCEs) endowed with biological activity, we have considered the possibility of using a novel combination approach to the BTZ scaffold to investigate its antimicrobial and anticancer properties)

Response: We have removed the antimicrobial activity part from the manuscript. Initially, it was there but later on, we highlighted only one key biological activity with in-depth discussion.

Point 11: Figure 6-7 shows that docking interactions are not perfect because some amino acids, which are part of the active site, present obstacles and lead to destabilization of the protein-ligand complex.

Response: We have modified the orientation of the ligand-protein complex to highlight the entry of the ligand on the protein active site. Now it is showing no blockade for entry into the binding site. It was observed through the docking energy and the combinations of interactions in the binding pocket, it was expected that the ligands have formed reasonably good interactions. Figures 6 & 7 images are replaced with the new improved quality images.

Point 12: The authors did not mention the nature of bleaching earth, whether it was used as is or in activated form. Did you regenerate after isolation of the catalyst (bleaching earth) for further use in the reaction?

Response: The active form of bleaching earth clay was used for the synthesis and it was regenerated after each cycle for use in the next cycle of reaction. We have tested the efficiency of bleaching earth clay for five runs with the concomitant activation after each run and was found to be as effective as the first run of application in the catalyzing properties.

Point 13: Check this sentence logically, “catalytic quantity of bleaching earth (10 moles, pH 12.5) (line; 304)” Do you have any idea when a catalytic amount/quantity is written or called?

Response: It was our inadvertent mistake. It is not the catalytic quantity. We have modified that statement. Specified quantity was used for the current method of synthesis. Thank you for providing useful comments for improving the manuscript further.

Round 2

Reviewer 1 Report

The authors took into account the comments of reviewers and significantly improved the manuscript. My decision is accept.

Author Response

Dear Editor

Manuscript: molecules-1740233

Thank you for reviewing our manuscript. We have incorporated the suggested changes in the manuscript. Here are our specific responses to the reviewers’ comments (Typed in BLUE font)

Thank you for your kind consideration.

Regards

Reviewer 2 Report

The article was improved, but still, it can't be published in Molecules in the present Form. English needs improvement. Chapter: The IR spectra should be overwritten. Authors should find and use a synonym of "due". English still need corrections.

Author Response

(The authors gave the same response as above.)

Reviewer 3 Report

Reason for reject paper, 

Author Response

(The authors gave the same response as above.)
